# ITS Efficiency Analysis for Multi-Target Tracking in a Clutter Environment

**Zvonko Radosavljević [1,*,†], Dejan Ivković [1,†] and Branko Kovačević [2]**

1   Millitary Technical Institute, 11030 Belgrade, Serbia; dejan.ivkovic@vti.vs.rs
2   School of Electrical Engineer, University of Belgrade, 11000 Belgrade, Serbia
*   Correspondence: zvonko.radosavljevic@gmail.com
†   These authors contributed equally to this work.

**Abstract:** The Integrated Track Splitting (ITS) is a multi-scan algorithm for target tracking in a cluttered environment. The ITS filter models each track as a set of mutually exclusive components, usually in the form of a Gaussian Mixture. The purpose of this research is to determine the limits of the 'endurance' of target tracking of the known ITS algorithm by analyzing the impact of target detection probability. The state estimate and the a-posteriori probability of component existence are computed recursively from the target existence probability, which may be used as a track quality measure for false track discrimination (FTD). The target existence probability is also calculated and used for track maintenance and track output. This article investigates the limits of the effectiveness of ITS multi-target tracking using the method of theoretical determination of the dependence of the measurements likelihood ratio on reliable detection and then practical experimental testing. Numerical simulations of the practical application of the proposed model were performed in various probabilities of target detection and dense clutter environments. Additionally, the effectiveness of the proposed algorithm in combination with filters for various types of maneuvers using Interacting Multiple Model ITS (IMMITS) algorithms was comparatively analyzed. The extensive numerical simulation (which assumes both straight and maneuvering targets) has shown which target tracking limits can be performed within different target detection probabilities and clutter densities. The simulations confirmed the derived theoretical limits of the tracking efficiency of the ITS algorithm up to a detection probability of 0.6, and compared to the IMMITS algorithm up to 0.4 in the case of target maneuvers and dense clutter environments.

**Keywords:** Data Association; Integrated Track Splitting; Interacting Multiple Models; Multi Targets Tracking

## 1. Introduction

In the surveillance space methodology, data arrives from the sensor in equal time intervals. The number of targets usually is a priori unknown. Each measurement has an unknown source (clutter or target). The tracks are initialized and updated using measurements, thus both true and false tracks exist at any time interval. Target measurements are only present in a scan with some probability of detection $P_D < 1$ [1]. A decision on the trajectory and existence of the target must be made exclusively based on the measurements received during one-time intervals (between successive measurements), regardless of the type of measurement source. Each track is initialized based on measurements from two adjacent time intervals and can be 'false' tracks or 'true' tracks. 'False' tracks are those that do not follow the target and 'true' tracks are those that follow the target [2].

To increase the target tracking efficiency, the false track discrimination (FTD) procedure is established. As the track quality measure, the Multiple Hypothesis Tracking (MHT) [3,4] performs the track score procedure, which is based on the Sequential Probability Ratio Test-SPRT. The false track rejection procedure is used to reject false tracks and (at the same time)

confirm true tracks [5,6]. After the first attempts to establish a procedure for measuring the quality of the track at MHT, the Probability Data Association (PDA) algorithm [7] was created. It includes the target observation, which appears in the generalized pseudo-Bayesian (GPB) PDA algorithm [8]. Also, the target detectability was established by the Interacting Multiple Model (IMM) PDA [9,10]. The target existence probability is first introduced in the literature, with the single target Integrated Probabilistic Data Association (IPDA) [6]. Most target tracking algorithms focus solely on estimating (linear or non-linear) measurements from sensors, often overlooking Data Association. This means that the task of assigning new measurements to existing tracks is typically determined by measuring the statistical distance between them [11]. In algorithms based on IPDA, track components are formed around the track, each with its parameters [12,13]. A set of components jointly participate in forming the track. The track components are Gaussian, so the track trajectory probability density function is the sum of the products of individual track components and their associated probabilities. By introducing the measurements likelihood ratio, it is possible to calculate the track existence probability through the Markov Chains model by the sequence of possible events associated with each new measurement to existing tracks [14–16]. At the same time, non-linear algorithms (the Unscented Kalman Filter (UKF), Bearings Only Tracking (BOT), Particle filter (PF), Generalized Labeled Multi-Bernoulli Tracking (GLMBT), etc.) are looking for solutions for the introduction of Data Association and track quality measure [17–21]. Also, the Poisson multi-Bernoulli mixture (PMBM) conjugates before multiple extended object filtering. A Poisson point process is used to describe the existence of yet undetected targets, while a multi-Bernoulli mixture describes the distribution of the targets that have been detected [22]. The particle filters [19] sample nonlinear (non-Gaussian) state probability density function (PDF) using a set of random particles [23]. They accommodate both nonlinear measurement and nonlinear state propagation. Tracking with the state-dependent probability of target detection is also proposed for the Gaussian Mixture Probability Hypothesis Density (GMPHD) filter [24]. However, the Data Association capabilities of the GMPHD filter appear to be significantly below the ITS-based trackers [25]. In previous research, the impact of $P_D$ on target tracking and Data Association quality was not significantly considered, especially outside the realm of ITS-based algorithms. In this paper, we will explore the influence of lower $P_D$ values at various clutter densities on tracking effectiveness. The contributions of this paper are as follows:

- A theoretical model of the dependence of the probability of target detection on the likelihood function (relative to the target existence probability) will be investigated for the well-known ITS algorithm, which has not been investigated in the literature so far;
- The theoretical results achieved should be proven practically, by numerical simulations, by obtaining probability of detection values that enable efficient tracking of a maneuvering target in a dense cluttered environment;
- To compare the obtained results, an efficient combined algorithm that successfully tracks maneuvering targets (IMMITS) was tested in parallel.

The rest of the paper is organized as follows: common theoretical assumptions and models used are presented in Section 2. The framework of the Integrated Track Splitting approach is detailed in Section 3, considering the impact of the probability of detection on the effectiveness of tracking (expressed through the probability of the track existence). A way to apply the Interacting Multiple Model algorithm to the ITS is presented in Section 4. The proposed approach is indicated by simulations in Section 5, followed by the concluding remarks in Section 6.

## 2. Assumptions and Models

Assume that a target exists and is always detectable with a given probability of detection $P_D$. Dynamic models of target trajectory are usually described by $\sigma_k^\tau$. The existence of a target being followed by each track is a random event, which is defined at each time interval $k$ for each track $\tau$:

- $\chi_k^\tau$ -the event that the track is following a target, i.e., that the target exists;
- $\bar{\chi}_k^\tau$ -the event that the track is not following a target, i.e., that the target does not exist;
- $\sigma_k^\tau = \mu$ -the event that the trajectory of the target $\tau$ uses model $\mu$ in the time interval between $k - 1$ and $k$.

where superscripts $\tau$ denote tracks. At each time interval, we initialize tracks by using random measurements of unknown origin, thus each track may be a true track (following a target) or a false track. Consider the existence $\chi_k^\tau$ at time $k$ of target $\tau$ as a random event, which is propagated as a Markov Chain process [6,11]. Also, each target follows a constant trajectory model between measurement times, and may switch to another model at each measurement time with a linear dynamic model [25,26]:

$$x_k^\tau = F_{(\sigma_k^\tau)} \cdot x_{k-1}^\tau + v_k^\tau(\sigma_k^\tau) \tag{1}$$

where $\sigma_k^\tau \in [1, M]$, $M$ is the number of trajectory propagation models, $F_{(\sigma_k^\tau)}$ is the state propagation matrix noise and the process noise $v_k^\tau$ is a zero mean and white Gaussian sequence with covariance matrix $Q_{\sigma_k^\tau}$.

### 2.1. Target Measurements

Each target $\tau$ generates one measurement $y_k$, at one sample interval. Due to the connecting path noise, the target comes with some probability of occurrence (probability of detection) $P_D(x_k^\tau)$. Consider the target trajectory state $x_k^\tau$. Then the linear measurement equation is given by [27]:

$$y_k^\tau = H x_k^\tau + \omega_k^\tau \tag{2}$$

where the additive measurement noise $\omega_k^\tau$ is a zero mean white Gaussian sequence with covariance matrix $R_k$.

### 2.2. Sensors

At each scan, the sensor returns a random number of target measurements and a random number of clutter measurements. The measurement of existing and detectable targets is taken with a given probability of detection. At time $k$, one sensor delivers a set of measurements $z_{k,j} = \left\{z_{k,j}\right\}_{j=1}^{M_k}$, out of which a set of measurements are selected for track update, where measurement $z_{k,j}$ denotes the $j - th$ element of $z_k$. The infinite resolution sensors are assumed, where each measurement has only one origin (either a target or the clutter) [28].

### 2.3. Clutter Measurements Model

Consider the nonhomogeneous clutter measurements as Poisson process density. At this time, the intensity of the Poisson process (at point $y$) in the measurement space can be denoted by $\rho(y)$ and is a priori known. Then the sensor measurements are estimated by $\rho_{k,j}(y)$ and its mean is $\rho(z_{k,j})$ [29]. The measurements are generated from one or more sensors. At time $k$, sensors deliver a set of measurements $z_{k,j}$, $j = 1, .., m_k$ out of which a set of measurements is selected for track update. The measurement sets may be empty, with $M_k = 0$. Denote by $Z^k$ the sequence of selected measurement sets up to and including time $k$, $Z^k = \{Z^{k-1}, z_{k,1}, z_{k,2}, ..., z_{k,j}, ..., z_{k,m_k}\}$. Measurements may originate from targets as well as from other objects. Otherwise, the sequence of measurements sets may be denoted by $Z^k = \left\{Z^{k-1}, z_k\right\}$ [30–32].

## 3. Integrated Track Splitting Filter Approach

The existing target tracking approach performs the FTD procedure at each scan and each track. To perform a false track discrimination procedure, ITS determines target existence based on the average track measurement likelihood ratio; thus, ITS is a mean

target existence estimator. We calculate the target existence probability $\psi_k$ and trajectory state PDF recursively. Relative to the value of $\psi_k$, we conclude the following:

- if the value of $\chi_k$ is less than the threshold, the target does not exist and we terminate the track,
- if the value of $\chi_k$ is greater than the threshold, the target exists and the track is confirmed.

*3.1. ITS Propagation*

Each track is a set of components, which is represented by the mean $\hat{x}^c_{k-1|k-1}$, covariance $P^c_{k-1|k-1}$ and probability, $\xi^c_{k-1}$, given that the track component measurement history $c$ is correct, and given that the target exists. Essentially, each track component represents a possible 'measurement-to-target' association history. Components are mutually exclusive.

The target trajectory state, at time interval $k$, is defined by a discrete event $\chi_k$ (target existence) and track trajectory state estimate $x_k$ [3,5]. In each iteration, a set of measurements, $z_k$, is selected in the current scan and is used to update the track state. Also, $Z^k = \{z_k; Z^{k-1}\}$ denoting the sequence of all selected measurements up to and including scan $k$. Within this terminology, we note that the a priori track state is conditioned on $Z^{k-1}$, and the a posteriori track state is conditioned on $Z^k$. To calculate both, a priori (propagation) and a posteriori (update) target existence probability, we use the Markov Chain One model. Then the propagation of the target existence probability from time $k-1$ to time $k$ is given by [7]:

$$P\{\chi_k|Z^{k-1}\} = \pi_{11}P\{\chi_{k-1}|Z^{k-1}\} \tag{3}$$

The value of $\pi_{11}$ depends on the time interval between measurements [6]. The track state estimates *PDF* of the single component is a single Gaussian *PDF*. The track component existence contains the target position at each measurement, assuming that the target exists. Thus, the a posteriori *PDF* of target trajectory state estimate at time $k-1$ is given by

$$p(x_{k-1}|\chi_k, Z^{k-1}) = \sum_{c=1}^{C_k} \xi^c_{k-1}p(x_{k-1}|\chi_k, c, Z^{k-1}) \tag{4}$$

where $\sum_{c=1}^{C_k}\xi^c_{k-1} = 1$, index $c$ denotes the track component, $\xi^c_{k-1}$ denotes the probability that track component measurement history $c$ exists, $C_k$ is the total number of components and $Z^{k-1}$ is the given measurement set, received before the current scan. At such time, we have predicted component state *PDF* as following [8]:

$$p(x_{k-1}|c, \chi_{k-1}, Z^{k-1}) = N(x_{k-1}; \hat{x}^c_{k-1|k-1}, P^c_{k-1|k-1}) \tag{5}$$

where $N$ represents Gaussian distribution. The *PDF* of the predicted target trajectory state is therefore a mixture of track component *PDF*s:

$$p(x_k|\chi_k, Z^{k-1}) = \sum_{c=1}^{C_k} \xi^c_{k-1}p(x_k|\chi_k, c, Z^{k-1}) \tag{6}$$

where

$$p(x_k|c, \chi_k, Z^{k-1}) = N(x_k; \hat{x}^c_{k|k-1}, P^c_{k|k-1}) \tag{7}$$

When the measurements arrive, each track component state *PDF* propagates (from $k-1$ to $k$), as a Kalman filter (KF) prediction by

$$\hat{x}^c_{k|k-1} = F\hat{x}^c_{k-1|k-1} \tag{8}$$

$$P^c_{k|k-1} = FP^c_{k-1|k-1}F^T + Q \tag{9}$$

where $F_k$ and $Q_k$ denote the state propagation and covariance matrix, between time intervals $k-1$ and $k$, respectively.

### 3.2. Measurements Selection

The selected measurements are those found within a region $V$ in measurement space, known as a *gate*, which is generally centered around the predicted measurement. The size of the selection gate depends on the measurements matrix $R$ and process noise matrix $Q$. Let $z_{k,j}$ denote the $j^{\text{th}}$ measurement in $z_k$. The gate size is based on the estimated PDF of the residual between the track and measurements. The gating region is defined as [9]:

$$d_{c,j}^2 = (z_{k,j} - \hat{z}_k^c)^T (S_k^c)^{-1} (z_{k,j} - \hat{z}_k^c) \tag{10}$$

The gate size is chosen so that all measurements $j$ satisfying $d_{c,j}^2 \leq (V)$ are retained; all others are discarded. The likelihood of each track component is given by

$$p_{k,j}^c = (1/P_G)N(z_{k,j}; \hat{z}_k^c, S_k^c) \tag{11}$$

$$z_k^c = H\hat{x}_{k|k-1}^c \tag{12}$$

$$S_k^c = HP_{k|k-1}^c H^T + R \tag{13}$$

Thus, the a priori measurements likelihood *PDF* for the measurement $z_{k,j}$ within the predefined gate is given by

$$p_{k,j} = \sum_{c=1}^{C_k} \xi_{k-1}^c p_{k,j}^c \tag{14}$$

### 3.3. ITS Update

In the update step, a component state estimate *PDF*, conditioned on the measurements component association history (assuming the target exists and is detected) is calculated. It is a Gaussian mixture of mutually exclusive component state *PDF*. The result is a track trajectory state estimate *PDF*. That means each track selects a set $z_k$ of $m_k$ candidate measurements. The probability of measurement selection in the gate is a parameter, defined as $P_G$. Its measurements likelihood, $p^c\{z_{k,j}|Z^{k-1}\}$, refers to measurement $z_{k,j}$, concerning tracking component $c$. Denote by $\rho_{k,j} \diamond \rho\{z_{k,j}\}$ clutter measurement density at $z_{k,j}$. Define measurement likelihood ratio at time $k$ by

$$\lambda_k = 1 - P_D P_G \sum_{j=1}^{m_k} \frac{p_{k,j}}{\rho_{k,j}} \tag{15}$$

Each measurement is paired with each old track component $c$ to create a new component. We obtain a state estimate of new components by the prediction of component state *PDF* $c$ paired with the measurement $z_{k,j}$. If the event $\chi_{k,j}$ that measurement outcome $j \geq 0$ is true, each pair denoted by $(c, j)$, where $j = 1, 2, ..., m_k$, generates a new component, either from an old component selected or from a 'null' measurement. Each new component $c^+$ is defined by

- probability of new component $\xi_k^{c_j^+}$,

- mean of new component $\hat{x}_{k|k}^{c_j^+}$, and

- covariance error of new component $P_{k|k}^{c_j^+}$

The a posteriori probability of component is given by

$$\xi_k^{c_j^+} = \frac{\xi_{k-1}^c}{\lambda_k} \begin{cases} 1 - P_D P_G, & j = 0, \text{ 'null-measurements'} \\ P_D P_G \frac{p_{k,j}}{\rho_{k,j}} & j \geq 0. \end{cases} \tag{16}$$

The Kalman filter update for each target component is given by

$$\hat{x}_{k|k}^{c_j^+} = \hat{x}_{k|k-1}^c + K^c(z_{k,j} - H\hat{x}_{k|k-1}^c) \tag{17}$$

$$P_{k|k}^{c_j^+} = [I - K^c H]P_{k|k-1}^c \tag{18}$$

where $K^c$ is the Kalman gain for component $c$, given by

$$K^c = P_{k|k-1}^c H^T (S_k^c)^{-1} \tag{19}$$

Finally, the updated target existence probability for the next ITS iteration is calculated by the following:

$$P\{\chi_k|Z^k\} = \frac{\lambda_k P\{\chi_k|Z^{k-1}\}}{1 - (1 - \lambda_k)P\{\chi_k|Z^{k-1}\}} \tag{20}$$

From the point of view of computational efficiency, it is proportional to the number of track component updates. Component termination, merging, and pruning techniques are used [5], along with the elimination of small probability components. We have two techniques to reduce the number of components.

*3.4. ITS Algorithm: Analysis of Effectiveness*

The effectiveness of the target tracking system depends on the parameters of the sensor, the connecting path (from the sensor to the tracking system) and the environment. The most significant parameters are the probability of target detection, clutter density and the reflectivity of the surface of a target. The reflectivity of the surface of the target often depends on the position of the target relative to the sensor, which can be attributed to the maneuver of the target. Therefore, we will limit ourselves to the influence of the target detection probability, the clutter measurement density and target maneuver. In the Prediction step, a priori target existence probability is calculated, by Equation (8). In the Update step, a posteriori target existence probability (for the next iteration) $\psi_{k|k}$ is calculated by the following:

$$\psi_{k|k} = P\{\chi_k|Z^k\} = \frac{\lambda_k P\{\chi_k|Z^{k-1}\}}{1-(1-\lambda_k)P\{\chi_k|Z^{k-1}\}} = \frac{\lambda_k \psi_{k|k-1}}{1-(1-\lambda_k)\psi_{k|k-1}} \tag{21}$$

Also, the likelihood is a function of the detection probability $P_D$ and some factor $\vartheta_k$, given by the following equation:

$$\lambda_k(P_D) = 1 - P_D P_G + P_D P_G \sum_{i=1}^{m_k} \frac{p_{k,i}}{\rho_{k,i}} = 1 - P_D \vartheta_k \tag{22}$$

where $P_G$ is the probability that the measurement belongs to the gate [4], and

$$\vartheta_k = P_G(1 - \sum_{i=1}^{m_k} \frac{p_{k,i}}{\rho_{k,i}}) \tag{23}$$

is independent of $P_D$. This iterative function is shown in Figure 1, for some selected frequent cases of the coefficient $\vartheta_k$. The track probability existence reaches its maximum for $P_D$ close to 1, while the $P_D \leq 0.4$, the probability of seeding a track tends to zero, so the algorithm below this value is unjustified.

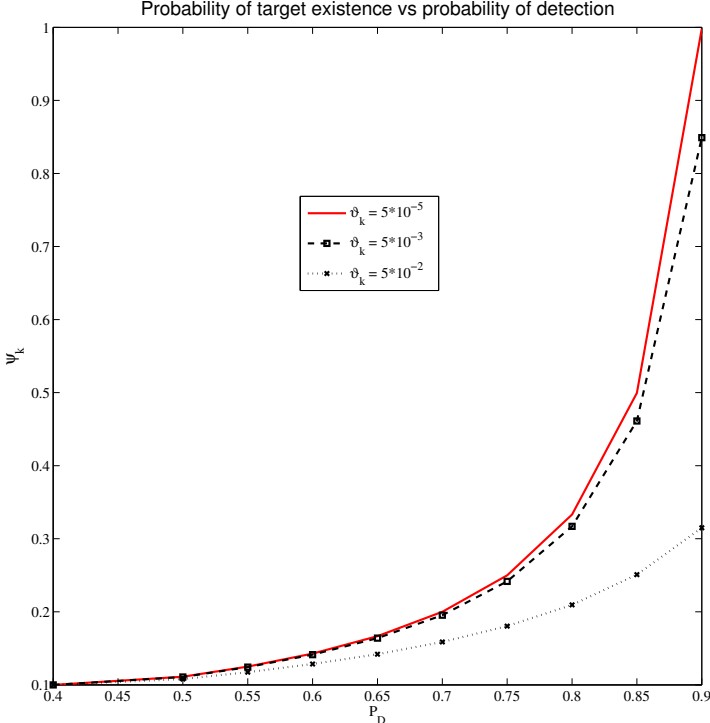

**Figure 1.** Target existence probability vs. probability of target detection diagram.

Based on the Prediction step, we have the a priori target existence probability:

$$\psi_{k|k-1} = \pi_{11}\psi_{k-1|k-1} \tag{24}$$

In the Update step, the *aposteriori* target existence probability is

$$\psi_{k|k}(P_D) = \frac{\lambda_k(P_D)\psi_{k|k-1}}{1 - (1 - \lambda_k(P_D))\psi_{k|k-1}} = \frac{\lambda_k(P_D)\pi_{11}\psi_{k-1|k-1}}{1 - (1 - \lambda_k(P_D))\pi_{11}\psi_{k-1|k-1}} \tag{25}$$

Additionally, we have

$$\psi_{k-1|k-1}(P_D) = \frac{\lambda_k(P_D)\pi_{11}\psi_{k-2|k-2}}{1 - (1 - \lambda_k(P_D))\pi_{11}\psi_{k-2|k-2}}. \tag{26}$$

By appropriate substitution, we have

$$\psi_{k|k}(P_D) = \frac{\lambda_k(P_D)\pi_{11}\frac{\lambda_k(P_D)\pi_{11}\psi_{k-2|k-2}}{1-(1-\lambda_k(P_D))\pi_{11}\psi_{k-2|k-2}}}{1 - (1 - \lambda_k(P_D))\pi_{11}\frac{\lambda_k(P_D)\pi_{11}\psi_{k-2|k-2}}{1-(1-\lambda_k(P_D))\pi_{11}\psi_{k-2|k-2}}} \tag{27}$$

By arranging the above equation, we get:

$$\psi_{k|k}(P_D) = \frac{\lambda_k(P_D)\lambda_{k-1}(P_D)\pi_{11}^2\psi_{k-2|k-2}}{1 - (1 - \lambda_k(P_D)\lambda_{k-1}(P_D)\pi_{11})\pi_{11}\psi_{k-2|k-2}} \tag{28}$$

Given that $\pi_{11} \approx 1$, and since the above equation is iterative, it can be completed (for the general case) as follows:

$$\psi_{k|k}(P_D) = \frac{\psi_{k-n|k-n}(P_D)\prod_{i=1}^{n}\lambda_{k-i}(P_D)}{1 - (1 - \prod_{i=1}^{n}\lambda_{k-i}(P_D))\psi_{k-n|k-n}(P_D)}, n = 1, ..., k-1 \tag{29}$$

$$\lambda_{k-i}(P_D) = 1 - P_D P_G(1 - \sum_{j=1}^{m_k}\frac{p_{k-i,j}}{\rho_{k-i,j}}), i = 0, 1, 2, ..., n \tag{30}$$

## 4. Interacting Multiple Model ITS Algorithm

In the Interacting Multiple Model approach, the target obeys one of a predefined set of models. Also, the motion of the target switches between models according to a Markov chain with a known transition probabilities matrix. If $N$ is the number of possible target dynamic models, and $M_{k,r}^c$ is the event that the target dynamic model at time $k$ was $r$, $r = 1, ..., N$, with component $c$ being the true component given that the target exists, then the component model probability can be defined as [9]:

$$\mu_{k|k-1,r}^c = P\{M_{k,r}^c|Z^{k-1}\} \tag{31}$$

Next, the PDF $p^c(x_k|Z^{k-1})$ of the component state estimate is defined as the sum of products *PDF* of the component model state estimate ($p_r^c(x_k|Z^{k-1})$) and the component model probabilities $\mu_{k|k-1,r}^c$, $r = 1, ..., N$ by the following equation:

$$p^c(x_k|Z^{k-1}) = \sum_{j=1}^{N} \mu_{k|k-1,j}^c p_j^c(x_k|Z^{k-1}) \tag{32}$$

The *PDF* of component measurements for all dynamic models is given by the following:

$$p^c(z|Z^{k-1}) = \sum_{r=1}^{N} \mu_{k|k-1}^{c,r} p_j^c(x_k|Z^{k-1}) \tag{33}$$

Each component *PDF* model state estimate $p_j^c(x_k|Z^{k-1})$ is described with its mean $\hat{x}_{k|k-1,r}^c$ and error covariance $S_{k|j}^c$; thus, we have

$$p_j^c(z|Z^{k-1}) = \frac{1}{P_G} N\{z, \hat{z}_{k|k-1,j}^c, S_{k|k-1,j}^c\} \tag{34}$$

where $\hat{z}_{k|k-1,j}^c$ is the predicted measurement for component $c$ and $S_{k|k-1}^c$ is the associated measurement error covariance matrix from the Kalman filter.

*Interacting Multiple Model ITS Approach*

The one recursion of IMMITS algorithms starts with the set of measurements from sensor $z_k$. State estimate *PDF*, $p_r^{c+}(x_k|Z^k)$, for the component $c+$ and model $r$ is obtained by applying measurement $z_{k,j}$ to $p_r^c(x_k|Z^{k-1})$. First, we update the model probabilities (from the previous time interval) and calculate the posterior probability that model $r$ is correct, given that component $c+$ is the true component, given by the following equation:

$$\mu_{k|k,r}^{c+} = \frac{\mu_{k|k-1,r}^c p_r^c(z_{k,j}|Z^{k-1})}{\sum_{r=1}^{N} \mu_{k|k-1,r}^c p_r^c(z_{k,j}|Z^{k-1})} \tag{35}$$

Next, we get the track state estimate and error covariance, for each component $n$, given by the following:

$$\hat{x}_{k|k}^{c+} = \sum_{r=1}^{N_r} \mu_{k|k,r}^{c+} \hat{x}_{k|k}^{c+} \tag{36}$$

$$P_{k|k}^{c+} = \sum_{r=1}^{N} \mu_{k|k,r}^{c+} [P_{k|k,r}^{c+} \hat{x}_{k|k}^{c+} (\hat{x}_{k|k}^{c+})^T] - \hat{x}_{k|k}^{c+} (\hat{x}_{k|k}^{c+})^T, \tag{37}$$

respectively. In the IMM mixing step, for each Markov model, the transition probabilities are calculated by the following:

$$\beta_{q|r} = P\{M_{k,r}^{c+}|M_{k-1,q}^{c+}\} \tag{38}$$

It produces the model prediction probabilities by the following equation:

$$\mu_{k+1|k,s}^{c+} = \sum_{r=1}^{N} \beta_{r|s} \mu_{k|k,r}^{c+}. \tag{39}$$

As a result, the mixing probabilities $\mu_{k+1|k+1,r,j}^{c+}$ are calculated as

$$\mu_{k+1|k,r,j}^{c+} = \sum_{r=1}^{N} \beta_{r|j} \mu_{k|k,r}^{c+}. \tag{40}$$

Finally, the mixed component model state estimate (mean and covariance) for the next iteration is given by

$$\hat{x}_{k|k}^{c+} = \sum_{r=1}^{N} \mu_{k+1|k,r,j}^{c+} \hat{x}_{k|k}^{c+,r} (\hat{x}_{k|k}^{c+,r})^T \tag{41}$$

$$P_{k|k}^{c+} = \sum_{r=1}^{N_r} \mu_{k+1|k,r,j}^{c+} [P_{k|k}^{c+,r} + \hat{x}_{k|k}^{c+,}(\hat{x}_{k|k}^{c+,r})^T] - \hat{x}_{k|k}^{c+,r}(\hat{x}_{k|k}^{c+,r})^T \tag{42}$$

respectively.

## 5. Simulations

The experimental scenario selected for numeric simulation analysis is the two-dimensional (positions and velocities), four-state aircraft tracking problem in which the sensor observes both position coordinates, assuming that they are independent. This area is $x = [0; 1000]$ [m] long and $y = [0; 1000]$ [m] wide. The clutter measurements satisfied a Poisson distribution. For programming the proposed theoretical models of algorithms, the MATLAB R2013a software package was used with Intel(R) Cote (TM)i5-4460 CPU@3.2 GHz and 4 Gb RAM. Two cases were examined, relative to the clutter density:

- Case 1: clutter density $\rho = 5 \cdot 10^{-5}$ [m$^{-2}$]
- Case 2: clutter density $\rho = 2 \cdot 10^{-4}$ [m$^{-2}$].

Both cases were examined with single target and multi-target scenarios. The main difference between the simulation scenarios is that the first scenario represents a single target with a straight-line trajectory scenario, while the second scenario represents multiple targets with a combined straight line and maneuver trajectory scenario. Both scenarios were tested in a Poisson clutter environment of densities $5 \cdot 10^{-5}$ [m$^{-2}$] and $2 \cdot 10^{-4}$ [m$^{-2}$]. The ITS parameters are calculated online according to the appropriate equations. The period of scanning is $T = 1$ s. For initialization of tracks, we use a *two point differencing* methodology [5]. The target dynamics are linear Gaussian models. The system is modeled as the vector state $x_k = [x \ \dot{x} \ y \ \dot{y}]$, where $x, y$ are the Cartesian coordinates of the target position, and $\dot{x} and \dot{y}$ are the appropriate velocities.

In the first simulation scenario, the single target moves in a straight line, at a constant speed, modeled by the constant velocity $F_{CV}$ transition matrix. In the second simulation scenario, two targets move alternately in straight lines and with maneuvers (left and right). The dynamic of the targets is modeled with the transition matrices, $F_{CV}$, $F_{CT}^L$ and $F_{CT}^R$, respectively, by the following:

$$\mathbf{F_{CV}} = \begin{pmatrix} 1 & T & 0 & 0 \\ 0 & 1 & 0 & 0 \\ 0 & 0 & 1 & T \\ 0 & 0 & 0 & 1 \end{pmatrix} \tag{43}$$

$$\mathbf{F_{CT}^L} = \begin{pmatrix} 1 & sin(\omega T)/\omega & 0 & [cos(\omega T)]/\omega \\ 0 & cos(\omega T) & 0 & -sin(\omega T) \\ 0 & -[cos(\omega T)]/\omega & 1 & sin(\omega T)/\omega \\ 0 & sin(\omega T) & 0 & cos(\omega T) \end{pmatrix} \tag{44}$$

$$\mathbf{F_{CT}^R} = \begin{pmatrix} 1 & sin(\omega T)/\omega & 0 & -[cos(\omega T)]/\omega \\ 0 & cos(\omega T) & 0 & sin(\omega T) \\ 0 & [cos(\omega T)]/\omega & 1 & sin(\omega T)/\omega \\ 0 & -sin(\omega T) & 0 & cos(\omega T) \end{pmatrix} \tag{45}$$

The process noise matrix is given by:

$$\mathbf{Q} = \begin{pmatrix} T^4/4 & T^3/2 & 0 & 0 \\ T^3/2 & T & 0 & 0 \\ 0 & 0 & T^4/4 & T^3/2 \\ 0 & 0 & T^3/2 & T \end{pmatrix} \tag{46}$$

where $q = 0.25^2$ is a maneuver coefficient. Probability of detection is variable in the range of $P_D = 0.4$–$0.9$ (single target scenario) and in the range of $P_D = 0.5$–$0.9$ (multi-target scenario). The measurements matrix and measurements noise covariance matrix are governed by the following:

$$\mathbf{H} = \begin{pmatrix} 1 & 0 & 0 & 0 \\ 0 & 0 & 1 & 0 \end{pmatrix} \tag{47}$$

$$\mathbf{R} = \begin{pmatrix} \sigma_x^2 & 0 \\ 0 & \sigma_y^2 \end{pmatrix}_, \tag{48}$$

respectively, where $\sigma_x^2 = \sigma_y^2 = 25$ [m$^2$]. All experiments were conducted over two types of two-dimensional scenarios. The total duration of both scenarios is 60 scans, repeating over 250 Monte Carlo simulations. In the region of surveillance, the radar sensor generates clutter according to the Poisson distribution in each scan. Thus, the average number of clutter measurements observed in each scan for clutter density $5 \cdot 10^{-5}$ [m$^{-2}$] is 20 measurements per scan and is 200 measurements per scan for clutter density $2 \cdot 10^{-4}$ [m$^{-2}$]. In the experiments, all tracks are initiated by the initial probability $P_{init} = 0.02$. When the track quality measure rises above the confirmation threshold, the track is thus confirmed by the probability of confirmation $P_c = 0.99$. When the track probability of existence falls below the termination threshold $P_t/3$, the track is terminated. The thresholds are determined experimentally to deliver approximately equal numbers of confirmed false track statistics. With the confirmed false tracks statistics, the success rate of confirmed true tracks is used to compare false track discrimination performance. Before the start of the simulations, it is necessary to set the same level of confirmed false track. At the beginning of each iteration, each initiated track is the false track. The track becomes a true track when the state estimate is sufficiently close to the true target state. The track remains true as long as it selects detections from the target.

All results are provided via confirmed true tracks (CTT) diagrams and root mean square error (RMSE) of position (the overall for both targets). Also, the numerical value of the processor time per one Monte Carlo run of the experiment (CPU) is presented in the appropriate tables.

### 5.1. Results of Single Target Scenario

In the single target scenario, the target moves in a straight line at a constant speed. This occurs in different clutter environments, represented by Case 1 and Case 2 (Figures 2 and 3), respectively.

The results of numerical simulations for a single target scenario (ITS algorithm) are given by the CTT and RMSE diagrams. Figures 4 and 5 are given CTT diagrams for clutter density $\rho = 5 \cdot 10^{-5}$ and $\rho = 2 \cdot 10^{-4}$, respectively, while Figures 6 and 7 show the RMSE diagrams for clutter density $\rho = 5 \cdot 10^{-5}$ and $\rho = 2 \cdot 10^{-4}$, respectively. The diagram shows a constant decrease in tracking efficiency as the detection probability decreases, in the range from 0.9 to 0.4 (for clutter density $5 \cdot 10^{-5}$) and in the range from 0.9 to 0.6 (for clutter

density $2 \cdot 10^{-4}$). It can be considered that if CTT diagram values fall below 0.5, target tracking is not effective. Additionally, large values of RMSE are considered ineffective for target tracking.

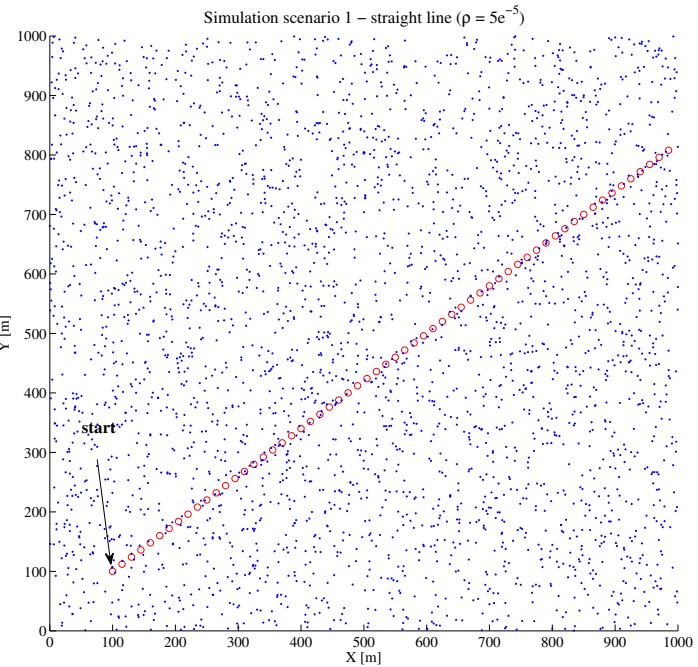

**Figure 2.** Single target scenario, $\rho = 5 \cdot 10^{-5}$.

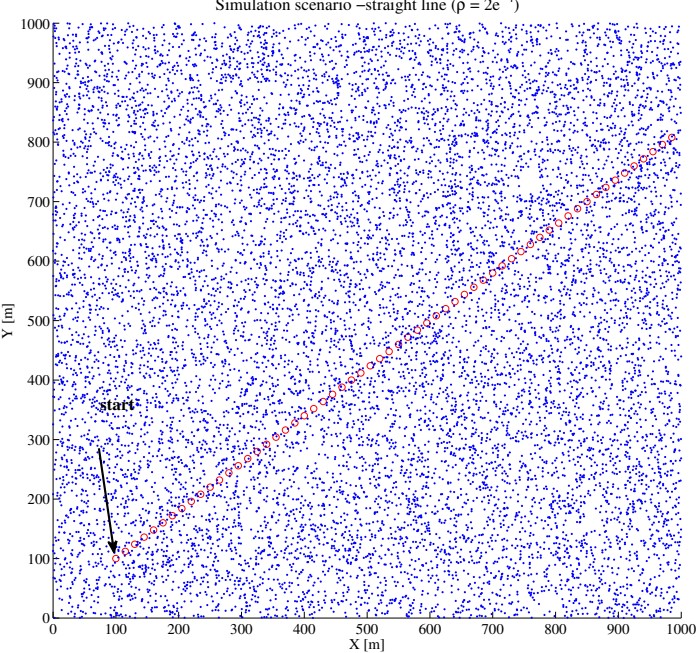

**Figure 3.** Single target scenario, $\rho = 2 \cdot 10^{-4}$.

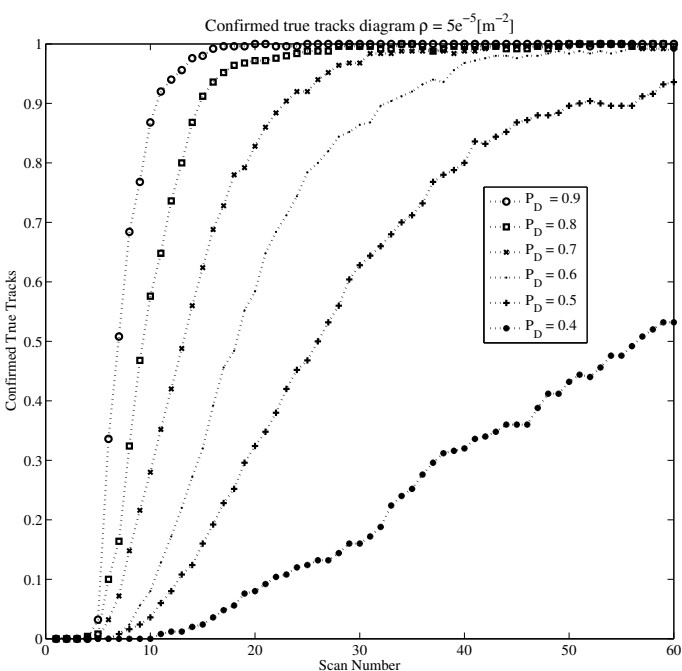

**Figure 4.** CTT diagrams for single target scenario, $\rho = 5 \cdot 10^{-5}$.

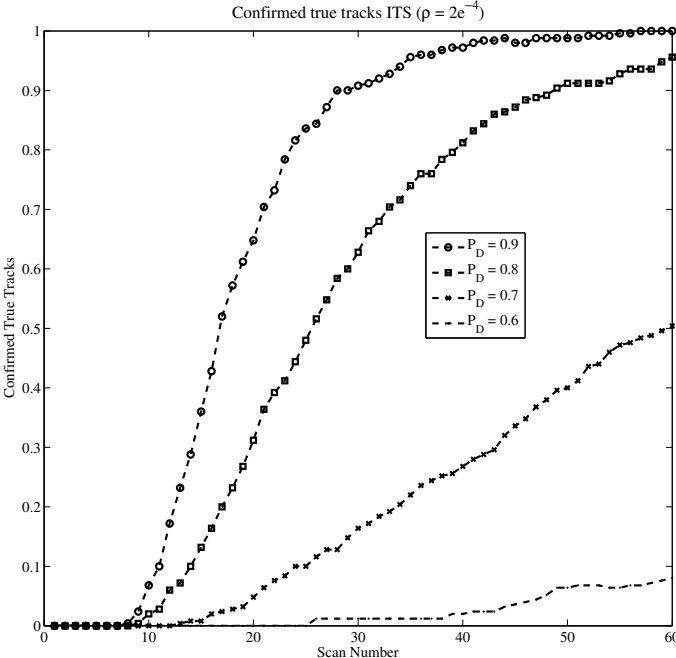

**Figure 5.** CTT diagrams for single target scenario, $\rho = 2 \cdot 10^{-4}$.

The results of the experiments (*CTT* and *RMSE* diagrams) clearly show the degradation of tracking quality, when $P_D$ decreases. In Case 1, tracking makes sense up to $P_D = 0.4$, while in Case 2, this limit is $P_D = 0.6$.

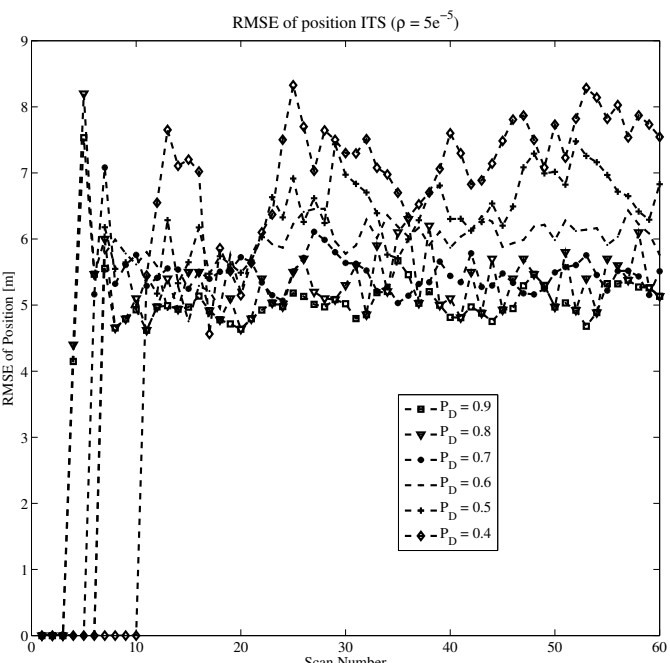

**Figure 6.** RMSE of position for single target scenario, $\rho = 5 \cdot 10^{-5}$.

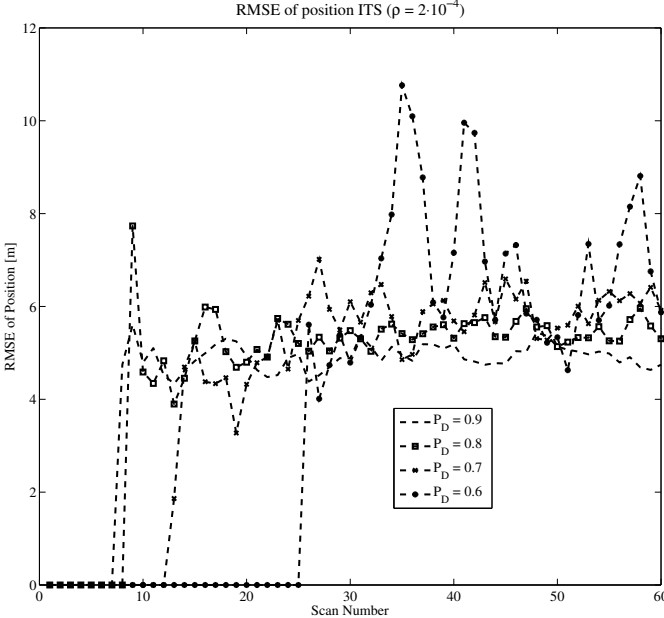

**Figure 7.** RMSE of position for single target scenario, $\rho = 2 \cdot 10^{-4}$.

### 5.2. Results of Multi-Target Scenario

The multi-target scenario, Figures 8 and 9 for Case 1 and Case 2, respectively, contains a combined movement of the target (straight line and maneuver). The initial states of the targets are $x_1 = [200; 14; 100; 10]'$ (target 1) and $x_2 = [100; 14; 800; -10]'$ (target 2). In this scenario, at the beginning, both targets move in a straight line at a constant speed towards the center of the surveillance region for fifteen scans, after which they enter a left maneuver (CT) with an angular speed $\omega = \pi/20$ for the next six scans. Then they continue in a straight line for eighteen scans and after that, enter the right maneuver (with the same angular speed) for about nine scans. In the last eleven scans of the simulation, the target moves in a straight line at a constant speed.

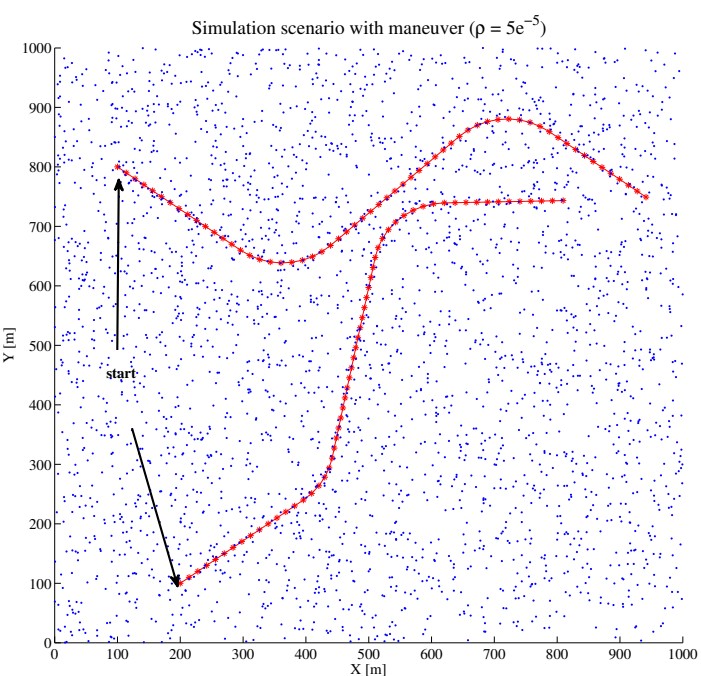

**Figure 8.** Multi target scenario, $\rho = 5 \cdot 10^{-5}$.

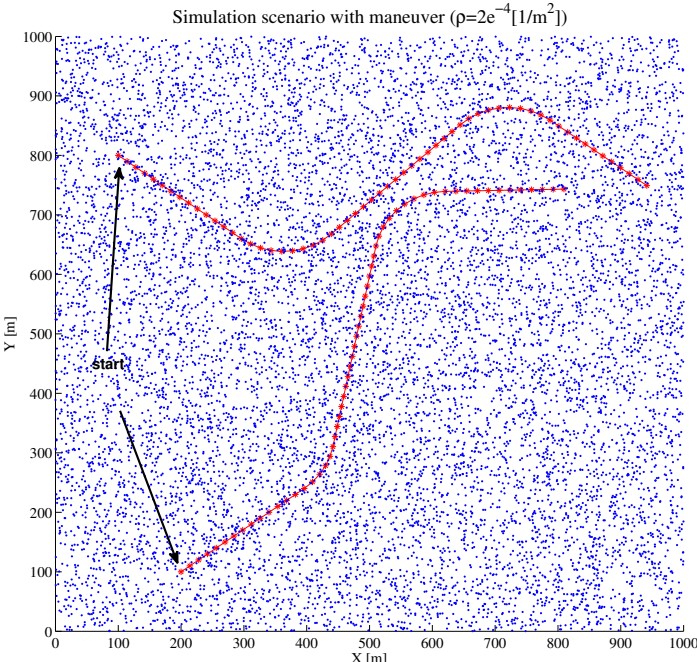

**Figure 9.** Multi target scenario, $\rho = 2 \cdot 10^{-4}$.

Also, the combined algorithm was obtained when the Integrated Track Splitting (ITS) tracker was extended by the incorporation of the Interacting Multiple Model (IMM) algorithm to enable the filter to efficiently track a maneuvering target in a various cluttered environment, named IMMITS. These are the uses of multi-scan tracking methods along with adaptive maneuver tracking.

The results of numerical simulations in a multi-target scenario are shown by the CTT diagrams, Figures 10 and 11 (ITS) for clutter density $\rho = 5 \cdot 10^{-5}$ [m$^{-2}$] and $\rho = 2 \cdot 10^{-4}$ [m$^{-2}$], respectively, while Figures 12 and 13 (IMMITS) represent clutter density $\rho = 5 \cdot 10^{-5}$ [m$^{-2}$] and $\rho = 2 \cdot 10^{-4}$ [m$^{-2}$], respectively. CTT diagrams show the decreasing efficiency of the ITS algorithm in the target maneuver, proportionally for all detection probabilities. At

$P_D = 0.6$, tracking is practically non-existent for both clutter densities. IMMITS shows significantly better tracking efficiency in the target maneuver, so it can track the target up to $P_D = 0.7$. Also, Figures 14 and 15 (ITS) display the *RMSE* of position diagrams for clutter density $\rho = 5 \cdot 10^{-5}$ [m$^{-2}$] and $\rho = 2 \cdot 10^{-4}$ [m$^{-2}$], respectively. Figures 16 and 17 display *RMSE* diagrams (IMMITS) for clutter density $\rho = 5 \cdot 10^{-5}$ [m$^{-2}$] and $\rho = 2 \cdot 10^{-4}$ [m$^{-2}$], respectively. Diagrams show a significant influence of the maneuver on the tracking efficiency. At the same clutter density and detection probability, the efficiency is lower by an order of magnitude in the case of the target's left and right maneuver.

The CPU time analysis, as depicted in Tables 1 and 2 for single target and multi-target scenarios respectively, indicates a higher consumption of processing time by the IMMITS algorithm compared to ITS. However, it also demonstrates the higher robustness of IMMITS in scenarios involving two maneuvering targets. Therefore, the use of the IMMITS algorithm is computationally justified in cases of dense clutter and maneuvering targets, while standard ITS can be retained in scenarios with weak clutter and straight-line target trajectories.

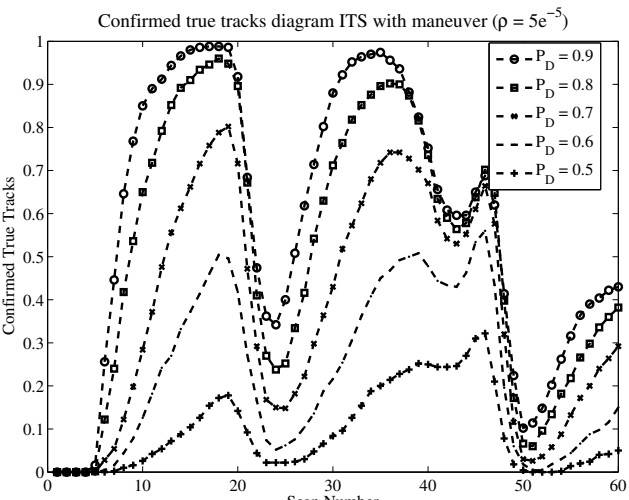

**Figure 10.** CTT diagrams for multi-target scenario (ITS) $\rho = 5 \cdot 10^{-5}$.

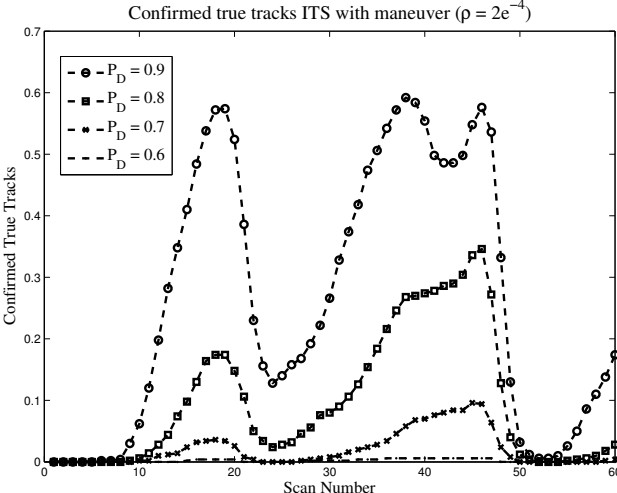

**Figure 11.** CTT diagrams for multi-target scenario (ITS), $\rho = 2 \cdot 10^{-4}$.

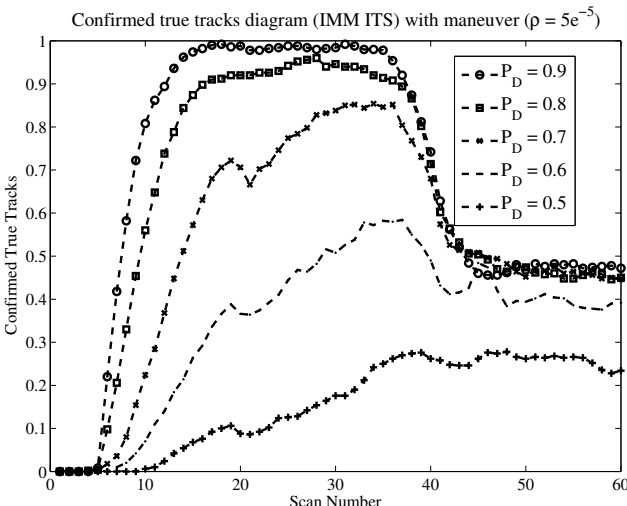

**Figure 12.** CTT diagrams for multi-target scenario (IMMITS), $\rho = 5 \cdot 10^{-5}$.

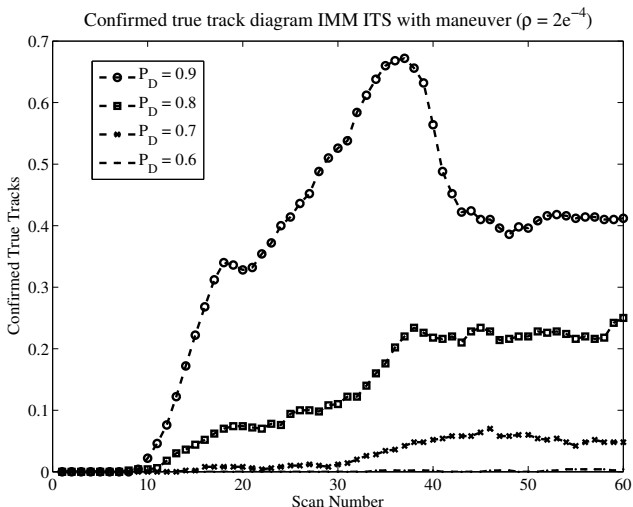

**Figure 13.** CTT diagrams for multi-target scenario (IMMITS), $\rho = 2 \cdot 10^{-4}$.

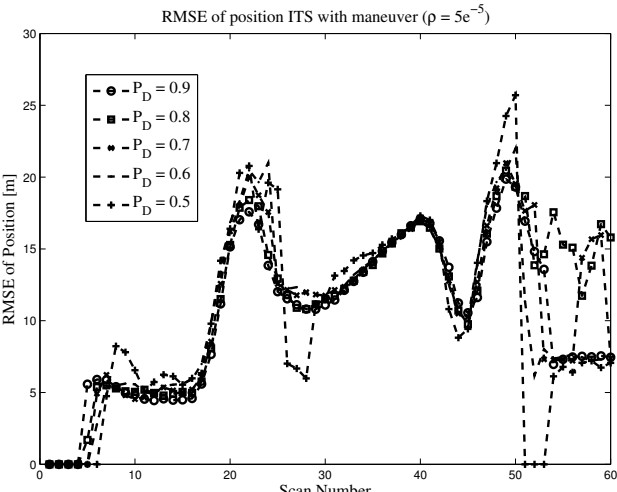

**Figure 14.** RMSE of position diagrams for multi-target scenario (ITS), $\rho = 5 \cdot 10^{-5}$.

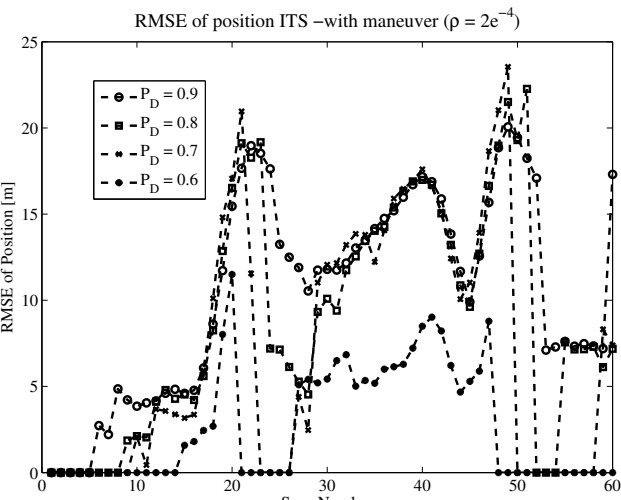

**Figure 15.** RMSE of position diagrams for multi-target scenario (ITS), $\rho = 2 \cdot 10^{-4}$.

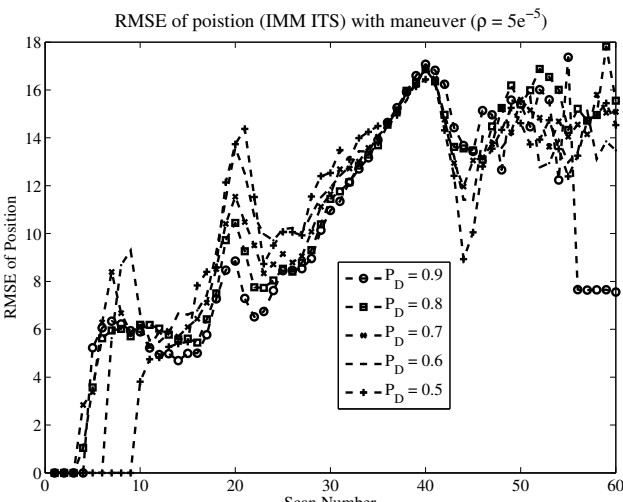

**Figure 16.** RMSE of position diagrams for multi-target scenario (IMMITS), $\rho = 5 \cdot 10^{-5}$.

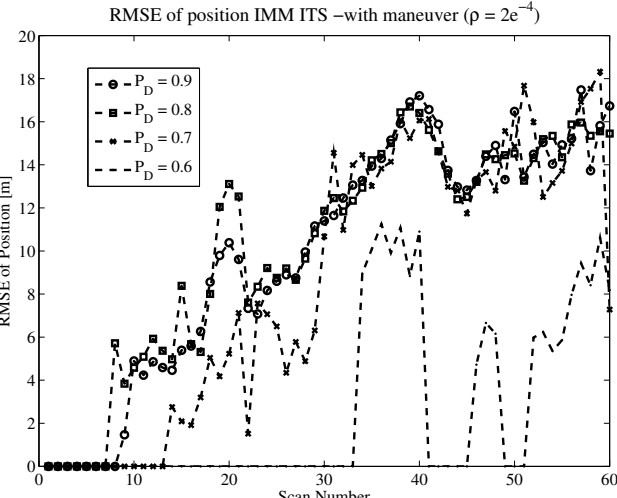

**Figure 17.** RMSE of position diagrams for multi-target scenario (IMMITS), $\rho = 2 \cdot 10^{-4}$.

In a multi-target scenario, the results of the experiments (CTT and RMSE diagrams) also show the degradation of tracking quality when $P_D$ decreases. In Case 1, tracking makes sense up to $P_D = 0.5$, while in Case 2, this limit is $P_D = 0.7$.

Tables 1 and 2 give results of CPU time for single and multi-target scenarios, respectively. From the diagrams, one can see the weak tracking efficiency, especially pronounced in the case of target maneuvers. In both tables, a slight increase in computing resources can be observed when reducing $P_D$.

The comparison table of the average RMSE of position for both single and multi-target scenarios, (Tables 3 and 4, respectively) shows an increase in tracking error during target maneuvers. A significant improvement is noticeable when using the IMMITS algorithm compared to the standard ITS. In other words, it is necessary to use IMMITS whenever maneuvering targets appear, especially when detection probabilities are low. However, it happens that the influence of heavy clutter and noise can briefly disrupt this relationship.

**Table 1.** CPU time for single target scenario (ITS vs. IMMITS).

| $\rho$ [m$^{-2}$] | $algor{\downarrow}P_D{\rightarrow}$ | 0.9 | 0.8 | 0.7 | 0.6 | 0.5 | 0.4 |
|---|---|---|---|---|---|---|---|
| $5 \cdot 10^{-5}$ | ITS | 1.60 | 1.82 | 1.89 | 2.31 | 2.61 | 2.83 |
| | IMMITS | 2.46 | 2.68 | 3.19 | 4.10 | 4.56 | 5.05 |
| $2 \cdot 10^{-4}$ | ITS | 14.8 | 16.2 | 17.0 | 17.4 | - | |
| | IMMITS | 24.8 | 28.2 | 32.3 | 35.1 | - | |

**Table 2.** CPU time for multi-target scenario (ITS vs. IMMITS).

| $\rho$ [m$^{-2}$] | $algor{\downarrow}P_D{\rightarrow}$ | 0.9 | 0.8 | 0.7 | 0.6 | 0.5 |
|---|---|---|---|---|---|---|
| $5 \cdot 10^{-5}$ | ITS | 1.84 | 2.06 | 2.23 | 2.7 | 3.0 |
| | IMMITS | 2.51 | 2.96 | 3.44 | 4.51 | 5.16 |
| $2 \cdot 10^{-4}$ | ITS | 15.9 | 17.2 | 17.8 | 19.1 | - |
| | IMMITS | 24.6 | 29.35 | 36.9 | 46.5 | - |

**Table 3.** Average RMSE [m] for single target scenario.

| $\rho$ [m$^{-2}$] | $algor{\downarrow}P_D{\rightarrow}$ | 0.9 | 0.8 | 0.7 | 0.6 | 0.5 | 0.4 |
|---|---|---|---|---|---|---|---|
| $5 \cdot 10^{-5}$ | ITS | 4.04 | 4.8 | 5.02 | 5.33 | 5.73 | 5.9 |
| | IMMITS | 4.88 | 5.28 | 5.94 | 6.31 | 6.56 | 6.68 |
| $2 \cdot 10^{-4}$ | ITS | 4.34 | 4.64 | 4.43 | 4.12 | - | - |
| | IMMITS | 4.98 | 5.49 | 5.93 | 3.09 | - | - |

**Table 4.** Average RMSE [m] for multi-target scenario.

| $\rho$ [m$^{-2}$] | $algor{\downarrow}P_D{\rightarrow}$ | 0.9 | 0.8 | 0.7 | 0.6 | 0.5 |
|---|---|---|---|---|---|---|
| $5 \cdot 10^{-5}$ | ITS | 10.04 | 11.04 | 11.0 | 10.6 | 10.1 |
| | IMMITS | 9.74 | 10.6 | 10.7 | 10.6. | 10.2 |
| $2 \cdot 10^{-4}$ | ITS | 10.5 | 8.96 | 7.24 | 2.67 | - |
| | IMMITS | 10.0 | 10.3 | 8.58 | 2.60 | - |

The advantages of IMMITS become apparent in the case of the target maneuver scenario, in which a significant advantage of IMMITS over the ITS algorithm is observed. Therefore, we suggest that future users use combined algorithms, especially in scenarios where stronger target maneuvers are anticipated. Overall, the numerical experiments showed that the ITS algorithm's 'endurance' limit for the single target scenario is $P_D = 0.4$ for clutter density $\rho = 5 \cdot 10^{-5}$ [m$^{-2}$] and $P_D = 0.5$ for the $\rho = 2 \cdot 10^{-4}$ [mm$^{-2}$] clutter density, while for the multi-target scenario with maneuvering targets is $P_D = 0.5$ for the clutter density $\rho = 5 \cdot 10^{-5}$ [m$^{-2}$] and $P_D = 0.6$ for the clutter density $\rho = 2 \cdot 10^{-4}$ [m$^{-2}$].

## 6. Conclusions

Analysis of target tracking efficiency on the known ITS algorithm was (theoretically and practically) examined in this paper. The results are the product of numerical experi-

ments for single and multiple targets, measuring track quality, and recursively calculating the track existence probability. The paper provides the testing of effectiveness for different clutter environments and the probability of detection. To compare the efficiency, the IMMITS algorithm is also given. The advantages of IMMITS become apparent only in the case of a multi-target scenario with multiple maneuvers, in which a significant advantage of IMMITS over the ITS algorithm is observed. The presented analysis and results provide the user with practical advice when choosing important parameters of the target tracking system.

The values of the parameters in which the algorithm can still work efficiently are defined; thus, the user can choose the optimal value of the tracking algorithm parameters. The lowest probability of detection that enables efficient tracking is $P_D = 0.4$, applicable to the typical clutter densities of $5 \cdot 10^{-5} \, [\mathrm{m}^{-2}]$ for the standard ITS algorithm. Extensive experiments showed that the dominant influence is the type of trajectory and the density of clutter, while the probability of detection is a less significant parameter.

**Author Contributions:** Conceptualization, Z.R. and D.I.; methodology, B.K.; software, Z.R.; validation, Z.R.; formal analysis, D.I.; investigation, D.I.; resources, Z.R.; data curation, D.I.; writing—original draft preparation, Z.R.; writing—review and editing, Z.R.; visualization, B.K.; supervision, B.K.; project administration, B.K. All authors have read and agreed to the published version of the manuscript.

**Funding:** This research was funded by the Ministry of Science, Technological Development and Innovations (Serbia), Contract No. 451-03-47/2023-01/200325.

**Data Availability Statement:** Data are unavailable due to privacy or ethical restrictions.

**Acknowledgments:** The authors would like to thank the Ministry of Science, Technological Development and Innovations (Serbia) and Ministry of Defense, Republic of Serbia. This work was supported by the Ministry of Science, Technological Development and Innovations (Serbia), Contract No. 451-03-47/2023-01/200325.

**Conflicts of Interest:** The authors declare no conflict of interest. The founders had no role in the design of the study; in the collection, analysis, or interpretation of data; in the writing of the manuscript; or in the decision to publish the results.

## Abbreviations

The following abbreviations are used in this manuscript:

| | |
|---|---|
| KF | Kalman Filter |
| UKF | Unscented Kalman Filter |
| EKF | Extended Kalman Filter |
| FTD | False Track Discrimination |
| GM | Gaussian Mixture |
| GMM | Gaussian Mixture measurements |
| GPB | Generalized Pseudo-Bayesian |
| PHD | Probability Hypotheses Density |
| PDF | Probability Density Function |
| MHT | Multiple Hypotheses Testing |
| MTT | Multi-Target Tracking |
| STT | Single Target Tracking |
| IMM | Interacting Multiple Models |
| ITS | Integrated Track Splitting |
| JITS | Joint Integrated Track Splitting |
| LM ITS | Linear Multi-target Integrated Track Splitting |
| PF | Particle Filter |
| IPF | Integrated Particle Filter |
| PDA | Probabilistic Data Association |
| IPDA | Integrated Probabilistic Data Association |

| FIST | Finite Sets Statistics |
| RSS | Random Set Statistics |
| SMC | Sequential Monte Carlo |
| JIPDA | Joint Integrated Probabilistic Data Association |
| SPRT | Sequential Probability Ratio test |

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
