# Peer review of "ITS Efficiency Analysis for Multi-Target Tracking in a Clutter Environment"

_remotesensing, doi:10.3390/rs16081471_

Round 1

Reviewer 1 Report

Comments and Suggestions for Authors

Paper can be interesting for a specific group of readers. I appreciate the effort to clarify the mathematical algorithm, despite some efforts could still be done to make the whole clearer also for non-specialized audience.

Performance improving of IMMITS in Confirming True Tracks is interesting, but it would be better detailed with some summary table as for RMSE. Lack of clear captions and clear description for Table 1 and 2 makes results understanding a bit difficult.

By the way, it seems that IMMITS leads to a higher RMSE: this drawback should be detailed better into Experimental Results or Conclusions. Moreover, it should be depicted which tradeoff must be considered for choosing between IMS and IMMITS in practical scenarios.

I think Analysis of Experimental Results and Conclusions paragraphs should be rewritten in a clearer way.

Comments on the Quality of English Language

I didnt get specific issues. Minor typos (i.e. "clatter").

Author Response

Thank you very much for taking the time to review this manuscript and to the useful suggestions. We hope that we will be able to respond correctly to your remarks and suggestions and that the overall manuscript will be of higher quality and content. Overall we expect that our research will provide useful advice to readers and that the selected journal will overall increase its rating. Please find the detailed responses below and the corresponding revisions/corrections highlighted in the re-submitted files.

Comments 1: Paper can be interesting for a specific group of readers. I appreciate the effort to clarify the mathematical algorithm, despite some efforts could still be done to make the whole clearer also for non-specialized audience. Performance improving of IMMITS in Confirming True Tracks is interesting, but it would be better detailed with some summary table as for RMSE.  

Response 1: Thank you for your suggestions. Table 3 and Table 4 have been added to the revised version (Page 19) and theirs comments to the Page 15 and Page 18:

The comparison table of the average RMSE of position for both single and multi target scenario, (Table 3 and Table 4, respectively) shows an increase in tracking error during target maneuvers. A significant improvement is noticeable when using the IMMITS algorithm compared to the standard ITS. In other words, it is necessary to use IMMITS whenever maneuvering targets appear, especially when for low detection probabilities. However, it happens that the influence of heavy clutter and noise can briefly disrupt this relationship

Comments 2: Lack of clear captions and clear description for Table 1 and 2 makes results understanding a bit difficult.

Response 2: Agree. We have, accordingly, revised caption to emphasize this point. New captions for Table 1 and Table 2 are given by the (Page 18): 

-           Table 1: CPU time for single target-straight line scenario (ITS vs IMMITS)

-           Table 2: CPU time for multi targets-maneuver scenario (ITS vs IMMITS)

CPU time analysis (Table 1 and Table 2) shows a higher consumption of processing time of the IMMITS algorithm compared to ITS, but also a higher robustness of IMMITS in the case of two maneuvering targets. This means that the use of the IMMITS algorithm is computationally justified in cases of dense clutter and maneuvering targets, while in the case of weak clutter and straight line target trajectories, the standard ITS can be retained.

Comments 3: By the way, it seems that IMMITS leads to a higher RMSE: this drawback should be detailed better into Experimental Results or Conclusions. Moreover, it should be depicted which tradeoff must be considered for choosing between IMS and IMMITS in practical scenarios. I think Analysis of Experimental Results and Conclusions paragraphs should be rewritten in a clearer way.

Response 3: Agree. The Results and Conclusions section are rewritten as:

-From the single target scenario (CTT and RMSE diagrams) one can notice the similarity or slight advantage of the ITS algorithm due to the fact that IMMITS makes a 'switch' from one dynamic model to another, while ITS always works with the CV dynamic model, which is aligned with the target's trajectory. The advantages of IMMITS become apparent only in the case of a multi-target scenario with multiple maneuvers, in which a significant advantage of IMMITS over the ITS algorithm is observed. Therefore, future users are suggested to use complex algorithms only in case of expected stronger target maneuvers.

Comments 4: I didnt get specific issues. Minor typos (i.e. "clatter").

Response 4: Thank you for your observations. Typo has been corrected.

4. Response to Comments on the Quality of English Language

Point 1:

Response 1:    Quality of English has been corrected.

5. Additional clarifications

Reviewer 2 Report

Comments and Suggestions for Authors

Abstract: It is not written appropriately. The abstract should be comprehensive, i.e. include purpose, methods and numerical findings.

Keywords have to be sorted alphabetically.

In the introduction, you should dedicate a paragraph for your proposed work and highlight what is your work is valuable

Figures are not clear, you have to make the details of the figures readable!

What type of simulator did you use? If Matlab, can you specify the parameters you have used?

What is the difference between scenario 1 and scenario 2?

Can you define IMMITS?

There is an ambiguity in defining scenario 1 and 2, can you please use figures?

Is CTT and IMMITS are metrics like RMSE? If yes, make a comparison between them, and highlight your outcomes.

You need to explain your results, it’s not preferable to allocate 3 lines to describe the results from Fig 4 –Fig 11 !

Reference list has to be updates, most of them are before 2006

Author Response

Thank you very much for taking the time to review this manuscript and to the useful suggestions. We hope that we will be able to respond correctly to your remarks and suggestions and that the overall manuscript will be of higher quality and content. Overall we expect that our research will provide useful advice to readers and that the selected journal will overall increase its rating. Please find the detailed responses below and the corresponding revisions/corrections highlighted in the re-submitted files.

Comments 1: Abstract: It is not written appropriately. The abstract should be comprehensive, i.e. include purpose, methods and numerical findings.  

Response 1: Thank you for your suggestions. Agree. We have, accordingly, revised abstract to emphasize this point. A purpose, methods and numerical findings are highlighted in the new abstract:

The Integrated Track Splitting (ITS) is a multi scan algorithm for target tracking in clutter environment. The ITS filter models each track as a set of mutually exclusive components, usually in the form of a Gaussian Mixture. The purpose of the research is to determine the limits of the 'endurance' of target tracking of the known ITS algorithm by analyzing the impact of target detection probability. The state estimate and the a-posteriori probability of components existence are computed recursively the target existence probability, which may be used as a track quality measure for the false track discrimination (FTD). The target existence probability, also calculated and used for track maintenance and track output. The article investigates limits effectiveness of ITS multi targets tracking, by the method of theoretical determination of the dependence of the measurements likelihood ratio on the reliable detection and then practical experimental testing. Numerical simulations of the practical application of the proposed model were performed in various probabilities of target detection and dense clutter environments. Additionally, the effectiveness of the proposed algorithm in combination with filters for various types of maneuvers using Interacting Multiple Model ITS (IMMITS) algorithms was comparatively analyzed. The extensive numerical simulation (which assume both straight and maneuvering targets), have shown up to which target tracking limits can be performed within different target detection probabilities and clutter densities. The simulations confirmed the derived theoretical limits of the tracking efficiency of the ITS algorithm up to a detection probability of 0.5, and compared to the IMMITS algorithm up to 0.4. in case of target maneuver and dense clutter environment.

Comments 2: Keywords have to be sorted alphabetically..

Response 2: Agree. New alphabetically keywords:

- Data Association; Integrated Track Splitting; Interacting Multiple Models; Multi Targets Tracking;

Comments 3: In the introduction, you should dedicate a paragraph for your proposed work and highlight what is your work is valuable

Response 3: Thank you very much for your suggestion. Agree. The Results and Conclusions section are rewritten. Was added in Introduction: 

- The contribution of the paper can be seen in:

- a theoretical model of the dependence of the probability of target detection on the likelihood function (relative to the target existence probability) will be investigated for the well-known ITS algorithm, which has not been investigated in the literature so far

- the theoretical results achieved should be proven practically, by numerical simulations, by obtaining probability of detection values that enable efficient tracking of a maneuvering target in a dense clutter environment

- in order to compare the obtained results, an efficient combined algorithm that successfully tracks maneuvering targets (IMMITS) was tested in parallel.

.

Comments 4: Figures are not clear; you have to make the details of the figures readable!

Response 4: Agree. Thank you for observations. All images have been converted into clearer formats (EPS) with a more comprehensive image description.

Comments 5: What type of simulator did you use? If MATLAB, can you specify the parameters you have used?

Response 5: For programming the proposed theoretical models the MATLAB software package was used on PC computer with Intel(R) Cote (TM)i5-4460 [email protected] GHz and 4 Gb RAM memory. Simulation parameters of proposed algorithm are given at the beginning of the Simulations Section.  

Comments 6: What is the difference between scenario 1 and scenario 2?

Response 6: The main difference between the simulation scenario1 and scenario 2 is that first scenario represents a single target with straight line trajectory, while second scenario represents multi targets with combined straight line and maneuver trajectory.  Both scenarios were tested in a two different Poisson clutter environment of densities 5*10^ {-5} and 2*10^ {-4} [1/m^2].

Comments 7: Can you define IMMITS?

Response 7: The  novel IMMITS algorithm it was obtained when known Integrated Track Splitting (ITS) tracker is extended by the incorporation of the Interacting Multiple Model (IMM) algorithm to enable the filter to efficiently track a maneuvering target in a various cluttered environment. The combination of the ITS filter for target tracking and the use of IMM filtering integrates two of the more powerful tools for target tracking. These are the use of multi-scan tracking methods along with adaptive maneuver tracking.

Comments 8: There is an ambiguity in defining scenario 1 and 2, can you please use figures?

Response 8: For clarity reasons, in the revised manuscript, scenario 1 (now named single target scenario) applied in clutter densities 5e^{-5} and 2e^{-4} , Fig 2, and Fig. 3, respectively and scenario 2 (now named multi-target scenario) applied in clutter densities 5e^{-5} and 2e^{-4}, Fig 8 and Fig.9. , respectively, which gives a total of 4 scenarios.

Comments 9: F Is CTT and IMMITS are metrics like RMSE? If yes, make a comparison between them, and highlight your outcomes.

Response 9: Only confirmed true tracks (CTT) and root mean square error (RMSE) are metrics, while IMMITS is the name of the algorithm.  

Comments 10: F You need to explain your results, it’s not preferable to allocate 3 lines to describe the results from Fig 4–Fig 11 !

Response 10: Agree. Thank you for your suggestion. A more detailed analysis of the results is provided in the revised manuscript (Page 10, 14, 15 and 18 ).

Comments 11: Reference list has to be updates; most of them are before 2006.

Response 11: Agree. Thank you for your suggestion. Reference list has to be updates; most of them are before 2006

- L. Zhuzheng, G. Bing and W. Jinfeng, "A Real-time Interactive Multi-Model (RT-IMM) Target Tracking Method," 2021 IEEE International Conference on Artificial Intelligence and Computer Applications (ICAICA), Dalian, China, 2021, pp. 503-507, doi: 10.1109/ICAICA52286.2021.9498215.

-G. Zhou, B. Zhu and X. Ye, "Switch-Constrained Multiple-Model Algorithm for Maneuvering Target Tracking," in IEEE Transactions on Aerospace and Electronic Systems, vol. 59, no. 4, pp. 4414-4433, Aug. 2023, doi: 10.1109/TAES.2023.3242944.

-Jiao, H. Liu, Y.,Yan, and J. Liu, H. A Refined Tracking Filtering Algorithm Based on IMM. 645-650. November 2023. doi: 10.1109/ICCAIS59597.2023.10382359.

4. Response to Comments on the Quality of English Language

Point 1:

Response 1:    Quality of English has been corrected.

5. Additional clarifications

Reviewer 3 Report

Comments and Suggestions for Authors

In the article, the authors present a performance analysis of an already published multi-target tracking algorithm, Integrated Track Splitting (ITS). The ITS algorithm has been completed with the inclusion of IMM filters in the trajectory tracking. The contribution of the article, according to the authors, is to develop an analysis of the performance of the IMM-ITS algorithm as a function of the sensor probability of detection (varying PD between 0.4 and 1).

Indeed, rigorous analysis of the influence of PD is not common. Although this parameter is usually varied between some values. The results of the article are interesting since they compare the performance of the IMM-ITS and the standard ITS algorithm.

However, in my opinion, the article should be improved in the following aspects before publication:

- In the introduction, explain more clearly what the contribution of the article is and its usefulness for the design of multi-target tracking algorithms.

- Improve the analysis of results and their discussion. There are no conclusions about what levels of PD are tolerable and under what conditions. It would be interesting to incorporate into the results a comparison with other classic algorithms of this type to serve as a reference level. It would be interesting to place the results of the different algorithms on the same graph or build comparative tables.

- Correct writing and text errors (disordered or incomplete sentences, incomplete acronyms in the bottoms of graphs...)

Comments on the Quality of English Language

Moderate editing of English language required

Author Response

Thank you very much for taking the time to review this manuscript and to the useful suggestions. We hope that we will be able to respond correctly to your remarks and suggestions and that the overall manuscript will be of higher quality and content. Overall we expect that our research will provide useful advice to readers and that the selected journal will overall increase its rating. Please find the detailed responses below and the corresponding revisions/corrections highlighted in the re-submitted files.

Comments 1: In the article, the authors present a performance analysis of an already published multi-target tracking algorithm, Integrated Track Splitting (ITS). The ITS algorithm has been completed with the inclusion of IMM filters in the trajectory tracking. The contribution of the article, according to the authors, is to develop an analysis of the performance of the IMM-ITS algorithm as a function of the sensor probability of detection (varying PD between 0.4 and 1). Indeed, rigorous analysis of the influence of PD is not common. Although, this parameter is usually varied between some values. The results of the article are interesting since they compare the performance of the IMM-ITS and the standard ITS algorithm. However, in my opinion, the article should be improved in the following aspects before publication:

- In the introduction, explain more clearly what the contribution of the article is and its usefulness for the design of multi-target tracking algorithms..  

Response 1: Thank you for your suggestions. Agree. In Section Introduction is added:

The contribution of the paper can be seen in:

- a theoretical model of the dependence of the probability of target detection on the likelihood function (relative to the target existence probability) will be investigated for the well-known ITS algorithm, which has not been investigated in the literature so far

- the theoretical results achieved should be proven practically, by numerical simulations, by obtaining probability of detection values that enable efficient tracking of a maneuvering target in a dense clutter environment

- in order to compare the obtained results, an efficient combined algorithm that successfully tracks maneuvering targets (IMMITS) was tested in parallel.

Comments 2: Improve the analysis of results and their discussion. There are no conclusions about what levels of PD are tolerable and under what conditions.

Response 2: Agree. We have, accordingly, revised Results and Conclusions Section to emphasize this point.

Was added in Conclusions:

Numerical experiments showed that the ITS algorithm's ‘endurance’ limit for the single target scenario is P_D=0.4 for the clutter density \rho=5*10{-5} and P_D=0.5 for the clutter density \rho=2*10{-4}, while for the multi target scenario with maneuvering targets it is P_D=0.5 for the clutter density \rho =5*10{-5) and P_D=0.6 for the clutter density \rho =2*10{-4}.

Comments 3: It would be interesting to incorporate into the results a comparison with other classic algorithms of this type to serve as a reference level.

Response 3: Thank you very much for your suggestion. Unfortunately, the competing classic algorithms for multi target tracking do not consider the probability of the track's existence, and do not have the durability of the ITS algorithm. In earlier research, the authors published a comparative analysis of the well-known Multi Hypothesis Testing (MHT) and ITS algorithms, in which MHT loses targets as early as 0.7. In earlier research, our scientific team was published (as a conference paper) the advantages of ITS algorithm in situations of low detection probability compared to the well-known multi target tracking algorithms MHT and PDA. Given that the publication of those results requires theoretical proofs in the mentioned algorithms (which would further increase the volume of the paper), the authors decided to publish these comparative analyzes in one of the following publications. As an example, the following figure shows a comparative analysis of ITS and MHT for P_D=0.7 multi target tvo straight line scenario, clutter density \rho=5*10{-5}. An additional experiment confirmed the superiority of ITS over the known MHT algorithm, which can be seen on the CTT diagram.

.

Comments 4: It would be interesting to place the results of the different algorithms on the same graph or build comparative tables.

Response 4:  Agree. Thank you for your suggestions Comparative new tables of ITS and IMMITS algorithm (Table 3 and Table 4) are given for single target and multi target scenarios and two densities (weak and heavy) of clutter. 

Comments 5: Correct writing and text errors (disordered or incomplete sentences, incomplete acronyms in the bottoms of graphs...)

Response 5: Agree. Thank you for suggestion. Corrections were made to writing, incomplete sentences and graphs.

4. Response to Comments on the Quality of English Language

Point 1:

Response 1: Quality of English has been corrected.

5. Additional clarifications

Reviewer 4 Report

Comments and Suggestions for Authors

The topic is interesting, and extended work for integrated track splitting filter is worth investigated. However, the innovation for this paper is limited. Authors should explain what is the difference of their work to e.g., 

D. Musicki, B. F. La Scala and R. J. Evans, "Integrated track splitting filter - efficient multi-scan single target tracking in clutter," in IEEE Transactions on Aerospace and Electronic Systems, vol. 43, no. 4, pp. 1409-1425, October 2007

since the methodology is pretty much the same. Also, the literature review needs heavily modification as it omits many recent developments in tracking domain, that are relevent to the proposed work, e.g., :

K. Granstrom, M. Fatemi, and L. Svensson, “Poisson multi-Bernoulli ¨ mixture conjugate prior for multiple extended target filtering,” IEEE Transactions on Aerospace and Electronic Systems, vol. 56, no. 1, pp. 208–225, 2019

Godsill, et al. "An adaptive and scalable multi-object tracker based on the non-homogeneous Poisson process." IEEE Transactions on Signal Processing 71 (2023): 105-120.

In addition, the cited literature of C. Hue [21] has errors in their derivations, so a more correct paper regarding the same topic can refer to:

Davey, Sam. "Probabilistic multihypothesis trackerwith an evolving poisson prior." IEEE Transactions on Aerospace and Electronic Systems 51.1 (2015): 747-759.

Godsill, et al. "An adaptive and scalable multi-object tracker based on the non-homogeneous Poisson process." IEEE Transactions on Signal Processing 71 (2023): 105-120.

Comments on the Quality of English Language

The typos are quite common, including in the abstract.

Author Response

Thank you very much for taking the time to review this manuscript and to the useful suggestions. We hope that we will be able to respond correctly to your remarks and suggestions and that the overall manuscript will be of higher quality and content. Overall we expect that our research will provide useful advice to readers and that the selected journal will overall increase its rating. Please find the detailed responses below and the corresponding revisions/corrections highlighted in the re-submitted files.

3. Point-by-point response to Comments and Suggestions for Authors

Comments 1: The topic is interesting, and extended work for integrated track splitting filter is worth investigated. However, the innovation for this paper is limited. Authors should explain what is the difference of their work to e.g.,

   D. Musicki, B. F. La Scala and R. J. Evans, "Integrated track splitting filter - efficient multi-scan single target tracking in clutter," in IEEE Transactions on Aerospace and Electronic Systems, vol. 43, no. 4, pp. 1409-1425, October 2007

since the methodology is pretty much the same.

Response 1: Thank you for your suggestions. We accept. The contribution (and the difference from the mentioned references) of the manuscript is also stated in the Introduction section, and it refers to the (so far) unexplored dependence of the measurement likelihood ratio (MLR) on the probability of target detection P_D. This dependence is directly reflected in the track existence probability, i.e. the false track discrimination procedure (FTD). In this way, the probability that a track will be lost from the zone of effective tracking can be calculated, theoretically. In Introduction is added:

‘’The contribution of the paper can be seen in:

- a theoretical model of the dependence of the probability of target detection on the likelihood function (relative to the target existence probability) will be investigated for the well-known ITS algorithm, which has not been investigated in the literature so far

- the theoretical results achieved should be proven practically, by numerical simulations, by obtaining probability of detection values that enable efficient tracking of a maneuvering target in a dense clutter environment

- in order to compare the obtained results, an efficient combined algorithm that successfully tracks maneuvering targets (IMMITS) was tested in parallel.’’

Comments 2: Also, the literature review needs heavily modification as it omits many recent developments in tracking domain, that are relevent to the proposed work, e.g., :

   K. Granstrom, M. Fatemi, and L. Svensson, “Poisson multi-Bernoulli ¨ mixture conjugate prior for multiple extended target filtering,” IEEE Transactions on Aerospace and Electronic Systems, vol. 56, no. 1, pp. 208–225, 2019

   Godsill, et al. "An adaptive and scalable multi-object tracker based on the non-homogeneous Poisson process." IEEE Transactions on Signal Processing 71 (2023): 105-120.

Response 2: Thanks for the helpful suggestion. Suggested references are listed and taken into consideration (section Introduction). 

  [21] Q. Li, R. Gan, J Liang, and S. Godsill, An Adaptive and Scalable Multi-object Tracker based on the Non-homogeneous Poisson Process, IEEE Trans. Signal Processing}, vol. 71, pp. 105-120, January 2023. DOI 10.1109/TSP.2023.3240498

 [22] K. Granstrom, M. Fatemi, and L. Svensson, Poisson multi-Bernoulli mixture conjugate prior for multiple extended target filtering, IEEE Transactions on Aerospace and Electronic Systems, vol. 56, no. 1, pp. 208–225, June 2019. DOI: 10.1109/TAES.2019.2920220.

Regarding to [22], it was added in Introduction:

’’Also, the Poisson multi-Bernoulli mixture (PMBM) conjugates before multiple extended object filtering. A Poisson point process is used to describe the existence of yet undetected targets, while a multi-Bernoulli mixture describes the distribution of the targets that have been detected [22]’’,

Comments 3: In addition, the cited literature of C. Hue [21] has errors in their derivations, so a more correct paper regarding the same topic can refer to:

  Davey, Sam. "Probabilistic multihypothesis trackerwith an evolving poisson prior." IEEE Transactions on Aerospace and Electronic Systems 51.1 (2015): 747-759.

  Godsill, et al. "An adaptive and scalable multi-object tracker based on the non-homogeneous Poisson process." IEEE Transactions on Signal Processing 71 (2023): 105-120.

Response 3: Thanks for the helpful suggestion. The proposed by reviewer literature (Godsill et al.  [21]) has been listed and processed in Introduction.   

Comments 4: Comments on the Quality of English Language

The typos are quite common, including in the abstract.

Response 4: We accept and proceeded.

4. Response to Comments on the Quality of English Language

Point 1:

Response 1: Quality of English has been corrected.

5. Additional clarifications

Round 2

Reviewer 2 Report

Comments and Suggestions for Authors

Authors responded to the comments. 

Reviewer 4 Report

Comments and Suggestions for Authors

I have no further comments

Comments on the Quality of English Language

I have no further comments